# Residential Greenspace Is Associated with Lower Levels of Depressive and Burnout Symptoms, and Higher Levels of Life Satisfaction: A Nationwide Population-Based Study in Sweden

**DOI:** 10.3390/ijerph19095668

**Published:** 2022-05-06

**Authors:** Yannick Klein, Petra Lindfors, Walter Osika, Linda L. Magnusson Hanson, Cecilia U. D. Stenfors

**Affiliations:** 1Department of Psychology, Stockholm University, 114 19 Stockholm, Sweden; yannick.klein@psychology.su.se (Y.K.); pls@psychology.su.se (P.L.); 2Center for Social Sustainability, Department of Neurobiology, Care Science and Society, Karolinska Institute, 141 83 Huddinge, Sweden; walter.osika@ki.se; 3Centre for Psychiatry Research, Department of Clinical Neuroscience, Karolinska Institute, and Stockholm Health Care Services, 113 64 Stockholm, Sweden; 4Stress Research Institute, 114 19 Stockholm, Sweden; linda.hanson@su.se

**Keywords:** green infrastructure, urban, nature, mental health, epidemiological studies, city-planning, sustainable environments

## Abstract

Population-based studies of individual-level residential greenspace and mental health outcomes are still limited. Thus, the present study investigates greenspace–mental health associations—including depressive symptoms, burnout symptoms, and life satisfaction—in a population-based sample of adults, the Swedish Longitudinal Occupational Survey of Health, in 2016 (*n* = 14,641). High-resolution land cover of greenspace and green–blue-space was assessed at 50, 100, 300 and 500 m buffers around residential addresses. Higher residential greenspace and green–blue-space were associated with lower levels of depressive and burnout symptoms among non-working individuals and with higher life satisfaction in the whole study population, after controlling for age, sex, individual income, and neighborhood socioeconomics. The immediate residential-surrounding environment (50 m) consistently showed the strongest associations with the outcomes. Having a partner was associated with better mental health outcomes and with having more residential greenspace, and adjusting for this rendered greenspace–health associations mostly statistically non-significant. In conclusion, higher levels of greenspace and green–blue-space in the immediate residential-surrounding environment were associated with better mental health outcomes in the present study, which contributes additional nuances to prior studies. The importance of residential greenspace for public health, urban planning, and development is discussed.

## 1. Introduction

Mental health disorders represent a major cause of the global burden of disease and are associated with vast individual and societal costs [1,2]. This means that such disorders are a major public health concern both globally and locally. In Sweden, mental health disorders are associated with, e.g., long-term sick leave, rehabilitation costs, early work exit, and productivity loss, in addition to individual suffering [3]. Preventive system-based approaches toward societal mental capital improvement are thus vital when designing interventions and developing policies [4]. The inclusion of mental health as part of the UN Sustainable Development Goals (SDGs) in terms of good health and well-being, side-by-side with sustainable cities and communities, underlines the importance of integrating health and environmental policies [5]. To design successful interventions, it is important to understand if and how different aspects of the environment are linked to mental health outcomes, covering both common mental health problems and positive aspects of mental health, including well-being.

Human health is constantly being influenced by interactions with the environment [6]. Specifically, in the past decades, experimental and observational research studies have accumulated on the association between nature contact—such as exposures to green (vegetation) and blue (open water bodies) environments and stimuli—and different aspects of mental health and well-being [7,8,9,10,11]. Such environmental exposures can thus constitute important modifiable risk factors for mental health.

Theories that have addressed the beneficial impact of nature contact on different aspects of mental well-being and health include, e.g., the biophilia hypothesis [12], stress reduction theory [13], and attention restoration theory [14]. 

In line with experimental and field studies, residential greenspace (RGS) exposure has also been linked to better mental health outcomes in epidemiological studies [15,16,17]. For example, RGS was associated with lower distress and higher life satisfaction in a fixed-effects analysis using panel data in the UK [15], and lower levels of depression, anxiety, and stress in a population-based survey in Wisconsin [16]. Furthermore, higher RGS assessed at a 500 m buffer was associated with a lower prevalence of major depression in a large cross-sectional study in the UK [17]. Moreover, RGS exposure has been found associated with lower stress in terms of biomarkers of allostatic load [18], cortisol levels, and self-reported symptoms [19]. In some previous studies, RGS exposure was associated with better mental health, especially in certain groups, including the elderly, or people who spend most of their time at home during the day [20,21]. 

Despite growing empirical support for mental health benefits of residential greenspace, some studies found no or mixed associations between RGS and mental health outcomes [7,22,23]. However, greenspace has often been assessed at a non-specific aggregate level such as general greenness at the neighborhood, district, or even city level. This results in uncertainty of the actual individual exposure to greenness and limits the accuracy of exposure as well as the conclusions which can be drawn from study results (see, e.g., Labib et al., 2020 [24], where related methodological aspects are further discussed, including the Modifiable Areal Unit Problem). The inconsistencies in prior study results may thus partly relate to methodological issues, including the use of greenspace exposure measures at a crude level and not at the individual address level, only accounting for publicly available greenspace and not all greenspace, and/or studying limited/small samples. Consequently, there is a need for population-based studies investigating higher quality greenspace measures at the individual level. That is, there is a need for population-based studies assessing the role of greenspace in the more immediate residential surrounding environment which individuals are more automatically exposed to on a daily basis, which are more independent of mobility and physical activity levels [24,25]. 

The aim of the present study is thus to investigate the role of individual RGS—encompassing all greenspace, public and private—at different egocentric Euclidean residential buffer zones in the immediate residential surrounding, up to 500 m (at 50 m, 100 m, 300 m, 500 m radiuses), for mental health outcomes, in a population-based sample of adults residing in Swedish urban areas. Mental health outcomes entail both negative and positive dimensions, including depressive symptoms, burnout symptoms, and life satisfaction. Firstly, it is hypothesized that more RGS exposure is associated with better mental health outcomes in terms of lower levels of depressive and burnout symptoms, as well as higher levels of life satisfaction. These associations are expected to be stronger among individuals that may spend more time at their residential location. Due to the automatic exposure to RGS in the immediate residence-surrounding environment, a second hypothesis is that RGS in the closest residential buffer zones (i.e., 50 m, 100 m) plays the greatest role in mental health outcomes. Moreover, prior research has also found blue space (i.e., natural spaces consisting of open waters, such as sea, lakes, rivers) to be associated with beneficial effects on mental health [26,27,28,29]. Thus, green and blue spaces are both regarded as natural spaces with health-beneficial potential. Accordingly, the residential land cover of green and blue space combined is also assessed as a global measure of residential natural space with the potential to enhance mental health in the present study. Therefore, the third hypothesis is that associations may be stronger between mental health outcomes and residential natural spaces when combining both green and blue spaces, rather than when studying greenspace in isolation.

## 2. Methods

### 2.1. Study Population and Design

The present cross-sectional study is based on the 2016 survey of the Swedish Longitudinal Occupational Survey of Health (SLOSH)—a nationwide population-based study. SLOSH comprises participants in the Swedish Work Environment Surveys (SWES 2003–2011) who in turn are sampled from the Labor Force Surveys (LFS) conducted biennially by Statistics Sweden [30]. In the LFS, a random sample of approximately 20,000 people is biennially drawn from the Swedish population, stratified by county, sex, citizenship, and inferred employment status, and asked to participate in the LFS. These individuals are then contacted by telephone, from among whom a subsample of gainfully employed people, 16–64 years of age, are sent self-report SWES questionnaires [30]. 

The study sample includes respondents to the SLOSH survey conducted in the spring of 2016 (*n* = 19,360, 51% response rate, out of which *n* = 13,572 were gainfully working, and *n* = 5788 were non-working) who resided in urban areas in non-rural municipalities, resulting in a final sample of 14,641 individuals with residential address information (Figure 1). Of these, *n* = 10,365 were gainfully working, and *n* = 4276 were non-working. The locality of residence in a rural municipality or outside urbanized areas was assessed according to The Swedish Association of Local Authorities and Regions (SALAR) classification of Swedish municipalities [31], and demographic statistical area unit (DeSO) classifications regarding locality mainly within versus outside urbanized areas by Statistics Sweden [32]. 

### 2.2. Variables

#### 2.2.1. Residential Greenspace and Green–Blue-Space Exposure Assessment

RGS and residential blue space (RBS) in 2016 were assessed using high resolution geographic and land-use data, measuring the amount of land covered (in square meters) by green and blue space around participants’ residences (addresses), within 50 m, 100 m, 300 m, and 500 m egocentric Euclidean buffer zones (Figure 2), across Sweden. Land use data are based on multiple layers of geographic data, including Sentinel-2 multispectral satellite imagery with 10 × 10 m resolution (from the European Space Agency). Then, pixels are classified as vegetated (green), mixed or non-vegetated (non-green), and further refined in the resolution of land cover classification of green and blue space through multiple additional layers of land cover mapping data from the Swedish Mapping, Cadastral, and Land Registration Authority, the Swedish Transport Administration, and the Swedish Board of Agriculture, on land covered by built space (incl. buildings, paved land cover, road and rail networks, and other built space), and open water bodies (for further information, see Statistics Sweden, 2019, report on greenspace and green areas in urban areas 2015 [33]). An example map of total greenspace in one of the greenest urban areas in Sweden is provided in the Appendix A. Thus, the resolution of the land cover data is at least 10 × 10 m, but will often be smaller than 10 × 10 m. Absolute RGS values in m^2^ for the respective buffer zones were converted into percentages of land covered by RGS for the respective buffer zones. An additional variable assessing the land covered by green and blue space (GBS) was also used, representing the proportion of land cover of RGS and RBS combined (i.e., summed values of RGS+RBS within the respective buffer zones).

#### 2.2.2. Mental Health Outcomes

Depressive symptoms were measured using the six-item Symptom Checklist core depression scale (SCL-CD6), measuring the intensity of symptoms during the past 2 weeks on a 5-point Likert scale (coded 0–4, *α* = 0.91). The scale is a valid unidimensional depression scale with particular suitability for large population surveys [34]. The sum of the six-item responses constitutes a depression score, indicating the level of depressive symptoms, ranging from 0–24. Scores of 10–11 indicate mild depressive symptoms, 12–15 indicate moderate depressive symptoms, and 16–24 indicate severe depressive symptoms, according to the International Classification of Diseases (ICD-10).

Burnout symptoms were measured with the Shirom-Melamed Burnout Questionnaire (SMBQ) subscale for emotional exhaustion and physical fatigue, consisting of 8 items (including one reverse-scored item) rated on a 7-point Likert scale (*α* = 0.92) [35,36,37,38]. The cluster of symptoms conceptualized as burnout was initially conceived in relation to the context of having a straining work situation. Scores on the SMBQ range from 1–7 with high scores indicating high levels of burnout. An index score was computed based on the mean across the items.

Life satisfaction was measured with a single item, derived from the Whitehall II study [39] and validated earlier in three large and independent samples [40]. Here, respondents were asked to rate on a 7-point Likert scale from low to high satisfaction how satisfied or dissatisfied they were with their overall lives. 

#### 2.2.3. Control Variables

Individual-level demographic variables including age, sex, and individual annual net income, were obtained through administrative national registers. Information on relationship status was obtained through the SLOSH survey, asking whether the respondent is single or married/cohabiting with a partner. Working status was also obtained via the SLOSH survey, where participants who are gainfully working at least 30% of full time respond to an in-work questionnaire and are categorized as gainfully working, while those who are not gainfully working, or are doing so less than 30% of full time, respond to the questionnaire for non-working individuals and are categorized as such. 

Neighborhood average income level was operationalized as the average annual net income of the adult population residing within a 500 m buffer zone of participants’ residential addresses. Income variables represent the annual net income in hundreds of Swedish Crowns (SEK) with EUR 1 equaling approximately SEK 9.36 in 2015 [41] and were converted into income quartiles. Annual net income was derived from the Swedish administrative register Longitudinal Integration database for Health Insurance and Labour market studies (LISA) [42], and includes different sources of income including salary, pension, social benefits, etc.

Locality and urbanity of residency were also considered in terms of municipality size and location: being located in a large/metropolitan city area, medium-sized city area, or small city; and being located in the regional center/central municipality versus in a suburb in a metropolitan or big city region, versus small-town location (according to the classification of Swedish municipalities by the Swedish Association of Local Authorities and Regions [31]. Location within the municipality of residency was also considered in terms of being located in the central part or not, via DeSO location classifications made by Statistics Sweden [32]. 

### 2.3. Statistical Analysis

Sequential linear regressions were used to assess the association between residential greenspace and the mental health outcomes, adjusting for a priori selected control variables [43,44]. The assumptions for linear regression were met [45]. Variables included in the regression models were residential greenspace (model step 1), age and sex (model step 2), average neighborhood income level (model step 3), individual income level (model step 4), and marital/relationship status (model step 5). Additional adjustments for the residential locality in terms of more or less urban and central locations were made but did not affect results further and were thus excluded. Potential moderation of associations depending on work status was assessed by testing the interaction effect between the predictor (RGS, and GBS) and work status on the outcomes. 

## 3. Results

Descriptive statistics of the study variables are displayed in Table 1. 

### 3.1. Residential Greenspace and Depressive Symptoms

Higher residential greenspace (RGS) was associated with lower levels of depressive symptoms in unadjusted models in close and far buffer zones (buffer 50 m *β* = −0.025, *p* = 0.002; 100 m *β* = −0.014, *p* = 0.084; 300 m *β* = −0.016, *p* = 0.056; 500 m *β* = −0.017, *p* = 0.045), and in the closest buffer zone after adjustments for age, sex, average neighborhood income, and individual income (buffer 50 m *β* = −0.017, *p* = 0.037). In fully adjusted models, adding relationship status, associations between RGS and depressive symptoms were statistically non-significant across all buffer zones (Table 2).

On average, gainfully working participants were 13.61 years younger, and had higher levels of depressive symptoms (*M* = 5.21, *SD* = 5.06) than non-working participants (*M* = 4.20, *SD* = 4.89; *t* = −11.144, *p <* 0.001). Further, there were significant RGS by work-status interactions on depressive symptoms in the closer RGS buffer zones (buffer 50 m *β* = 0.031, *p* = 0.042, 100 m *β* = 0.036, *p* = 0.020). Stratifying for work status, RGS associations with depressive symptoms were stronger in the non-working subsample. Here, RGS was significantly associated with lower levels of depressive symptoms in unadjusted models across buffer zones (buffer 50 m *β* = −0.056, *p* < 0.001; 100 m *β* = −0.048, *p* = 0.002; 300 m *β* = −0.041, *p* = 0.008; 500 m *β* = −0.040, *p* = 0.011) and after adjustment for control variables (model 4), except after adjusting for relationship status (Table 3). RGS had no significant associations with depressive symptoms in the working subsample (Figure 3; Appendix A). 

Associations between GBS and depressive symptoms were stronger than associations between greenspace alone and depressive symptoms, across models 1–4, in the whole sample, as well as in the non-working sub-sample. Further investigating the non-working subsample, the association between GBS and lower levels of depressive symptoms was statistically significant in the widest buffer zone (buffer 500 m *β* = −0.034, *p* = 0.026), even in model 5 (Appendix A).

### 3.2. Residential Greenspace and Burnout Symptoms

Only in the closest buffer zone around the residence, RGS was significantly associated with lower levels of burnout symptoms in the unadjusted model (buffer 50 m β = −0.019, *p* = 0.023). The effect vanished after adjustments for age and sex, as well as after adjusting for average neighborhood income, individual income, or in fully adjusted models, adding relationship status (Table 4).

In general, gainfully working participants had higher average levels of burnout symptoms (M = 2.54, SD = 1.24) when compared to non-working participants (M = 2.15, SD = 1.23; *t* = −17.504, *p* < 0.001). Moreover, there were significant RGS by work–status interactions (buffer 50 m *β* = 0.029, *p* = 0.055; 100 m *β* = 0.033, *p* = 0.032; 300 m *β* = 0.035, *p* = 0.024; 500 m *β* = 0.032, *p* = 0.036). After stratification for working status, RGS associations with burnout symptoms were stronger in the non-working group. Here, RGS was significantly associated with fewer burnout symptoms in unadjusted models (buffer 50 m *β* = −0.046, *p* = 0.003, buffer 100 m *β* = −0.040, *p* = 0.011; 300 m *β* = −0.042, *p* = 0.005; 500 m *β* = −0.041, *p* = 0.008). After adjusting for control variables (model 4), more RGS was still significantly associated with lower levels of burnout symptoms in the wider buffer zones (300 m *β* = −0.032, *p* = 0.032; 500 m *β* = −0.032, *p* = 0.032). RGS associations were non-significant when including relationship status into the model (Table 5). Further, RGS was not significantly associated with burnout symptoms in the working subsample (Figure 4; Appendix A).

Running sequential regression models with GBS as exposure variable in the whole sample, associations with lower levels of burnout symptoms were stronger and statistically significant in unadjusted models (buffer 50 m *β* = −0.021, *p* = 0.011; 100 m *β* = −0.015, *p* = 0.069; 300 m *β* = −0.020, *p* = 0.017; 500 m *β* = −0.023, *p* = 0.006), compared to models with RGS exposure. In the non-working sub-sample, associations between GBS and burnout symptoms were stronger across models and buffer zones, and remained marginally significant in the wider buffer zone (buffer 500 m β = −0.029, *p* = 0.054), also in the fully adjusted model 5.

### 3.3. Residential Greenspace and Life Satisfaction

RGS was significantly associated with higher levels of life satisfaction across buffer zones and after adjustments for age, sex, individual income, and neighborhood income (buffer 50 m *β* = 0.032, *p* < 0.001; 100 m *β* = 0.025, *p* = 0.003; 300 m *β* = 0.024, *p* = 0.004; 500 m *β* = 0.018, *p* = 0.034). In fully adjusted models, adding relationship status, associations between RGS and life satisfaction were statistically non-significant across all buffer zones (Table 6).

On average, non-working participants had higher levels of life satisfaction (*M* = 5.92, *SD* = 1.25) than gainfully employed participants (*M* = 5.77, *SD* = 1.23, *t* = 6.32, *p* < 0.001). RGS associations with higher levels of life satisfaction were significant in the non-working group in unadjusted models (buffer 50 m *β* = 0.052, *p* = 0.001; 100 m *β* = 0.042, *p* = 0.007; 300 m *β* = 0.037, *p* = 0.018; 500 m *β* = 0.028, *p* = 0.072), as well as after adjustment for age, sex, individual income, and neighborhood income, across buffer zones (Table 7). In the working group, RGS was associated with higher levels of life satisfaction in unadjusted models, too, except for the 500 m buffer (Figure 5; buffer 50 m *β* = 0.025, *p* = 0.012; 100 m *β* = 0.019, *p* = 0.053; 300 m *β* = 0.020, *p* = 0.043; 500 m *β* = 0.014, *p* = 0.149). After adjusting for the same four variables as in the non-working group, RGS associations with higher levels of life satisfaction were significant in all buffer zones except for the 500 m buffer, in the working group (buffer 50 m *β* = 0.026, *p* = 0.008; 100 m *β* = 0.019, *p* = 0.047; 300 m *β* = 0.021, *p* = 0.032; 500 m *β* = 0.017, *p* = 0.085). After additional adjustment for relationship status, RGS associations with life satisfaction were non-significant in both the working and non-working groups (Table 8).

Models with GBS exposure showed similar association trends with life satisfaction as RGS models, with slightly elevated standardized coefficients compared to RGS models in the whole sample, as well as in the working and non-working group (Appendix A).

## 4. Discussion

### 4.1. General Discussion

Investigating associations between individual residential greenspace, within different egocentric Euclidean buffer zones, and mental health outcomes in a population-based adult sample, this cross-sectional study found that higher levels of greenspace primarily closely around the residence were associated with better mental health outcomes.

Specifically, higher levels of RGS exposure were consistently associated with lower levels of depressive symptoms, especially in the immediate environment around the residence (50 m buffer) and especially for adults who tend to spend more time at home (i.e., non-working), in line with the first two hypotheses. These effects remained even after adjusting for age, sex, average neighborhood income, and individual income. Associations were stronger when considering residential natural space encompassing green and blue space, compared to RGS only, in line with the third hypothesis. Concerning burnout symptoms, the pattern of RGS associations with the outcome was also in line with the first two hypotheses, with negative associations in the unadjusted models being strongest in the closest buffer zone around the residence (50 m buffer) and among non-working participants. As with depressive symptoms, the trending associations between burnout and GBS combined were stronger than for RGS only. However, in general, RGS and GBS associations with burnout symptoms in all models and buffer zones were weaker and less consistent after adjusting for confounders, compared to exposure associations with depressive symptoms. Important differences between the concepts and measures of depression versus burnout used in the current study include that the former concerns core depressive symptoms with a focus on affective symptoms of low mood, while burnout mainly concerns symptoms of physical, mental, and emotional fatigue. The differences in the exposure associations with depression versus burnout may indicate that RGS has a greater impact on the affective/emotional components of mental health, compared with fatigue and energy components of mental health.

Examining a positive mental health outcome, higher RGS was associated with higher levels of life satisfaction in both the working and non-working groups, and associations were particularly strong in the closest buffer zone, again supporting the first two hypotheses. Further, associations were stronger in models with GBS as an exposure variable, in line with the third hypothesis.

The findings of stronger associations between GBS combined and mental health outcomes are consistent with prior research on restorative qualities and benefits to mental health of both green and blue environments [26,27,28,29]. When assessing either type of environment separately, this may underestimate the amount of land covered by “natural spaces” that can be beneficial to health. For example, there may be plenty of natural space with benefits to mental health aspects even if greenspace is low if blue space is instead available. Likewise, there may be plenty of restorative natural environment in terms of greenspace but no blue space available. The results are in line with the third hypothesis that combining green and blue spaces can provide a fuller exposure assessment of the amount of land covered by natural spaces (green and blue) that can have beneficial effects on mental health and may therefore be more strongly related to mental health.

Having a partner (married or living together) was also associated with having more RGS, as well as better mental health outcomes. After adjustment for this factor, associations between greenspace and mental health did generally not remain statistically significant, except for associations between green–blue space (500 m buffer) and lower levels of depressive and burnout symptoms among non-working individuals.

### 4.2. Residential-Surrounding Greenspace

In the present study, the immediate environmental surroundings of individual residences were assessed, utilizing high-resolution land cover data, thus investigating the environment to which participants are exposed automatically on a daily basis when spending time at home (i.e., exposure does not depend on individuals actively seeking out a green or blue area some distance from the home), as this may have the most beneficial effects on mental health. This is in line with recent recommendations [24]. Neglecting the immediate residence-surrounding environment, excluding domestic gardens, or basing analyses on public greenspace accessibility only, would thus exclude the exposure that people have by default daily at home. Using measures of greenspace at a broader level (i.e., greenspace measures of the neighborhood, borough, district, or city of residence, rather than based on individual residential address), using only larger buffer zones or only publicly available greenspace are common limitations in prior research. Such limitations may partly explain the varying results of prior research, where some studies observed no beneficial associations between RGS and mental health outcomes or mixed results [7,22,23]. Using individual-level measures and small buffer zones, the risk of unassessed confounder impact, such as physical activity, is thus minimized [25]. These risks increase when studying larger buffer zones [25]. In the present study, the availability of all greenspace as close as the 50 m radius around the home address was studied in a population-based sample, thus contributing to empirical findings on RGS and mental health at a more fine-grained level, which has been relatively limited in prior research.

Green and blue space in the immediate residential-surrounding environment may exert beneficial effects, and facilitate micro-restorative experiences daily, which can enhance mental well-being and health by several means [11]. These include, e.g., visibility effects, including views from the home [46], or other sensual pathways through which greenspace and blue space may beneficially act upon affective and general well-being and stress reduction [46,47], such as nature sounds [48] or odors [49]. Moreover, nature contact can also facilitate other processes that are central to mental health—in particular stress-related symptoms and depression—including decreased rumination [50] and improved attention and executive cognitive performance [8,9,14,51], as well as an increased sense of connectedness and relatedness to the natural world [52,53]. Furthermore, window views from home have been found to be associated with higher well-being, too [9,46], possibly through micro-restorative experiences counterbalancing mental fatigue. Specifically, window views at home have been described as safe opportunities which allow for prospect, exploration, and fascination without danger [9].

Although the effects in the present study were small, the results underscore the importance of RGS proximity and that the green infrastructure should be present immediately surrounding people’s homes. Importantly, this greenspace is what people will be exposed to automatically, daily, which will support mental health in the population.

### 4.3. Context and Implications

When considering publicly available small coherent greenspaces (0.5 ha or more), access to residential coherent greenspace is generally good in Sweden, with 94% of urban residents having access to greenspace within 200 m of the home in 2015 [33]. On the other hand, the amount of greenspace can vary greatly between different urban areas and a rather high proportion (37%) of total greenspace is made up of private residential gardens and other spaces with restricted access [33].

Importantly, while most have access to at least one small greenspace within 200 m, we still find that the amount of general greenspace and green–blue-space particularly in the immediate surroundings of the home is associated with better mental health. However, other factors that were associated both with higher residential greenspace and better mental health were younger age, sex (being male), higher individual and neighborhood income level, and in particular relationship status. Those being married or living with a partner had both a greener residential environment and better mental health—findings that contribute additional nuances to prior research. After adjustment for this factor, associations between greenspace and mental health were generally not statistically significant, except for associations between green–blue space (500 m buffer) and lower levels of depressive and burnout symptoms among non-working individuals. While few studies have controlled for marital/partner status, White, Alcock, Wheeler, and Depledge (2013) [15] is an exception, where greenspace–health associations remained after adjustment for marital status. Such differences in study findings may again be explained by the use of different greenspace measurement methods. In White et al. (2013) [15], more crude, aggregate area-level greenspace measures were used, not measuring individual-level greenspace around the residence, as was done in the present study.

In line with prior empirical work [20,21], the results of the present study showed that residential greenspace was more important among those who may spend more time at home (i.e., who were currently not gainfully working). A majority, but not all, in the non-working group had reached retirement age and were therefore no longer gainfully working. However, as teleworking from home has increased since the COVID-19 pandemic (with ca. 30–40% teleworking at least partly during the pandemic [54]), residential greenspace may also have become increasingly important for those individuals who spend more time working from home, during and after the pandemic. This is an aspect for future research to address further since the present study concerns a time before the COVID-19 pandemic.

### 4.4. Strengths and Limitations

Strengths of the present study include the investigation of an adult nationwide population-based study sample and the assessment of objective high-resolution RGS and RBS at the individual level, and at multiple buffer zones around participants’ addresses. Moreover, validated self-report mental health measures were used, multiple control variables were considered, and subgroups of gainfully working versus non-working adults were investigated.

However, the use of self-report health scales can have both advantages and disadvantages. One possible disadvantage is that they may be subject to biased response sets (i.e., over- or under-reporting symptoms) or context effects. On the other hand, such biases in symptom reporting apply also in clinical settings and assessments. The major advantages of the self-report measures used in the present research are that they allow the investigation of symptom *levels,* which was the aim of the present study. That is, the full range of symptoms from none to severe are assessed, thus including sub-clinical symptoms, and not merely, e.g., a registered diagnosis. Furthermore, mental health is inherently about the individual experience (of positive and negative symptoms), whether reported in a de-identified survey or to a health care practitioner. It has also been found that people are good at rating their own general health, where such self-ratings are predictive of a range of health outcomes (all-cause mortality, future functional status, treatment outcomes), and more so than some “objective” health measures such as diagnoses [55]. Furthermore, in the present study, fine-grained individual objective environmental and demographic data were studied in relation to individual health measures, as such preventing risks of common method bias which can be an issue when both predictors and outcomes are self-rated [56]. However, further complementary studies of other clinical outcomes would be valuable. Generally, self-report measures with multiple items, such as the depression and burnout scales, yield better scale properties [57]. Here, the use of a single-item measure of global life satisfaction may be considered a limitation. However, single-item measures of life satisfaction have been found reliable, robust, and valid, for instance in terms of test-retest reliability and in producing similar results to those of multi-item scales [58], while minimizing the respondent burden, reducing criterion contamination, or improving face validity [57,59]. Furthermore, items asking about general satisfaction with life (as used in the present study) have been found to be the most reliable. This motivates the use of a single-item life satisfaction measure, which has also been used and validated before [39,40], to include a positive measure of mental health.

The present study focused on the role of individual residential greenspace exposure in the closer residential surrounding—exposure which is more automatic and less dependent on the individual’s mobility to reach green areas further away from the residence. However, this can also be regarded as a limitation. Non-residential activity- and mobility-dependent greenspace exposure in relation to health outcomes in population-based samples is an important area to study further, and examples of such methodologies already exist [24,60].

Further limitations include the inability to draw conclusions regarding causality due to the observational and cross-sectional design.

## 5. Conclusions

In consideration of the UN SDGs, in particular aiming to establish good health and well-being for all, as well as to build sustainable cities and communities, the present study delivers insights on the importance of the immediate residential-surrounding environment for key mental health outcomes, underscoring the need to prioritize and incorporate urban green infrastructure in the immediate residential living environment as part of sustainable urban planning and development. This seems to be particularly important for those individuals, who spend more time at home, which also has important implications for those in the workforce today who spend more time at home due to teleworking, since the COVID-19 pandemic.

## Figures and Tables

**Figure 1 ijerph-19-05668-f001:**
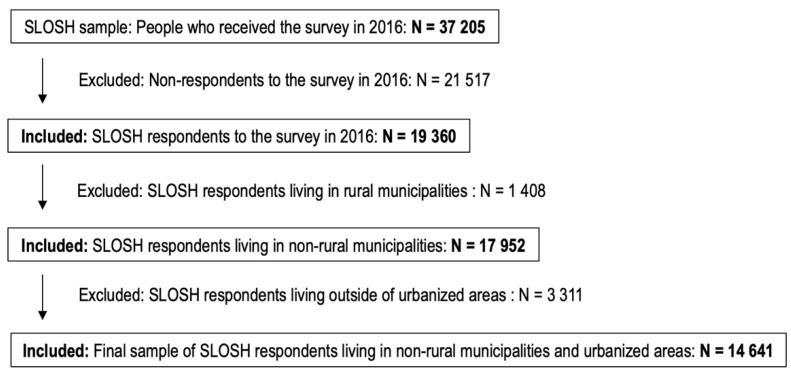
Participant sample, inclusion criteria.

**Figure 2 ijerph-19-05668-f002:**
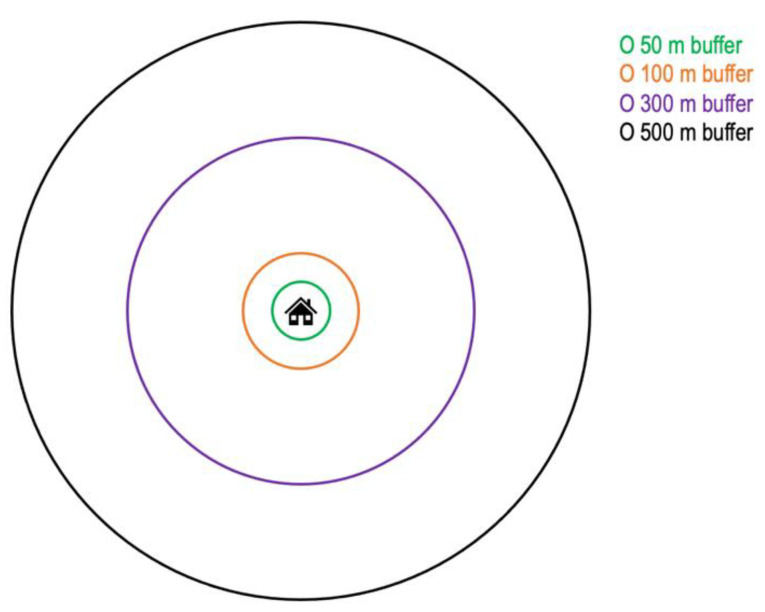
Residential greenspace and green-blue-space exposure assessment within 50 m, 100 m, 300 m, and 500 m egocentric Euclidean buffer zones.

**Figure 3 ijerph-19-05668-f003:**
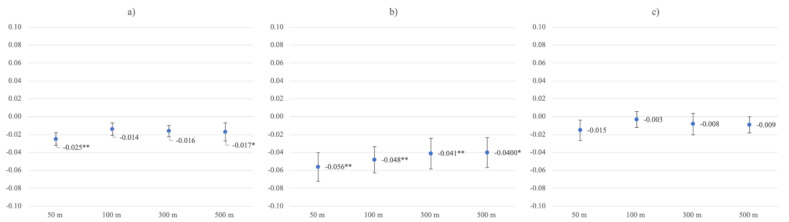
Greenspace ß coefficients with standard errors for unadjusted depressive symptoms models in the (**a**) whole sample, (**b**) non-working group, and (**c**) working group. * Statistical significance at *p* ≤ 0.05. ** Statistical significance at *p* ≤ 0.01.

**Figure 4 ijerph-19-05668-f004:**
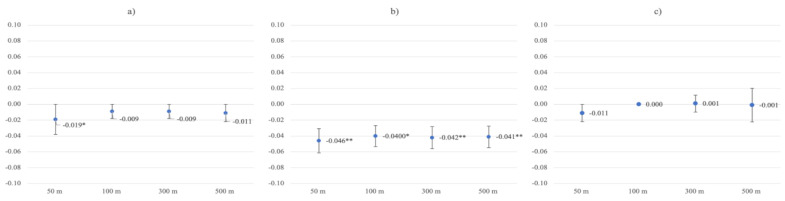
Greenspace ß coefficients with standard errors for unadjusted burnout symptoms models in the (**a**) whole sample, (**b**) non-working group, and (**c**) working group. * Statistical significance at *p* ≤ 0.05. ** Statistical significance at *p* ≤ 0.01.

**Figure 5 ijerph-19-05668-f005:**
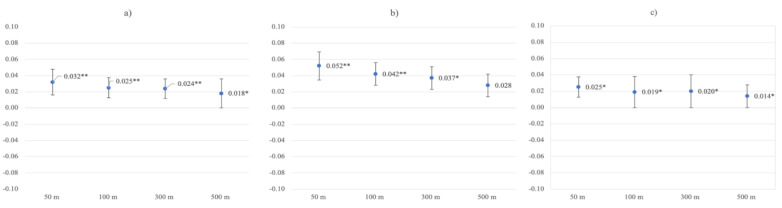
Greenspace ß coefficients with standard errors for unadjusted life satisfaction models in the (**a**) whole sample, (**b**) non-working group, and (**c**) working group. * Statistical significance at *p* ≤ 0.05. ** Statistical significance at *p* ≤ 0.01.

**Table 1 ijerph-19-05668-t001:** Descriptive statistics, mental health outcomes, RGS, GBS, and demographics.

Variable	Total Sample (*n* = 14,641)	Working (*n* = 10,365)	Non-Working (*n* = 4276)
Mean depressive symptoms score (SD)	4.91 (5.03)	5.21 (5.06)	4.20 (4.89)
Mean burnout score (SD)	2.43 (1.25)	2.54 (1.24)	2.15 (1.23)
Mean life satisfaction score (SD)	5.82 (1.24)	5.77 (1.23)	5.92 (1.25)
Mean RGS in percent			
50 m buffer zone (SD) 100 m buffer zone (SD) 300 m buffer zone (SD) 500 m buffer zone (SD)	54.08 (19.40) 56.01 (18.43) 56.82 (17.00) 56.13 (16.40)	54.24 (19.40) 56.14 (18.53) 56.93 (17.09) 56.22 (16.47)	53.70 (19.39) 55.69 (18.23) 56.54 (16.79) 55.92 (16.22)
Mean GBS in percent			
50 m buffer zone (SD) 100 m buffer zone (SD) 300 m buffer zone (SD) 500 m buffer zone (SD)	54.31 (19.25) 56.68 (18.12) 59.20 (16.30) 60.05 (15.50)	54.43 (19.27) 56.75 (18.21) 59.23 (16.39) 60.08 (15.58)	54.00 (19.20) 56.52 (17.89) 59.11 (16.07) 59.98 (15.29)
Count (%) Mean average neighborhood income in SEK			
Low income, <= 2205 Lower medium income, 2206–2495 Higher medium income, 2496–2908 High income, 2909+	2760 (18.9) 3517 (24.0) 3975 (27.1) 4390 (30.0)	1837 (17.7) 2440 (23.5) 2833 (27.3) 3255 (31.4)	923 (21.6) 1077 (25.2) 1141 (26.7) 1135 (26.5)
Count (%) Mean individual income in 100’s SEK			
Low income, <= 2175 Lower medium income, 2176–2818 Higher medium income, 2819–3617 High income, 3618 +	3116 (21.3) 3341 (22.8) 3757 (25.7) 4427 (30.2)	1022 (9.9) 2505 (24.2) 3103 (29.9) 3735 (36.0)	2094 (49.0) 836 (19.6) 654 (15.3) 692 (16.2)
Mean age	55.40 (11.70)	51.43 (9.76)	65.04 (10.35)
Count women (%)	8303 (56.7)	5 921 (57.1)	2382 (55.7)
Count men (%)	6338 (43.3)	4444 (42.9)	1894 (44.3)
Count married or cohabiting (%)	11,207 (76.5)	8025 (77.4)	3182 (74.4)
Count Single (%)	3232 (22.1)	2238 (21.6)	994 (23.2)

Note: Income variables represent the annual net income in hundreds of Swedish Crowns (SEK) with EUR 1 equaling approximately SEK 9.36 in 2015 [41].

**Table 2 ijerph-19-05668-t002:** Associations between residential greenspace at 50 m, 100 m, 300 m, and 500 m buffers, and depressive symptoms (whole sample/population).

	50 m Buffer				100 m Buffer				300 m Buffer				500 m Buffer			
Model, N = 14,290	β	B	Lower, upper (95% CI of B)	*p*	β	B	Lower, upper (95% CI of B)	*p*	β	B	Lower, upper (95% CI of B)	*p*	β	B	Lower, upper (95% CI of B)	*p*
1	−0.025	−0.007	−0.011, −0.002	0.002	−0.014	−0.004	−0.008, 0.001	0.084	−0.016	−0.005	−0.010, 0.000	0.056	−0.017	−0.005	−0.010, 0.000	0.045
2	−0.016	−0.004	−0.008, 0.000	0.045	−0.008	−0.002	−0.006, 0.002	0.353	−0.012	−0.004	−0.008, 0.001	0.132	−0.013	−0.004	−0.009, 0.001	0.104
3	−0.018	−0.005	−0.009, −0.000	0.032	−0.008	−0.002	−0.007, −0.002	0.317	−0.013	−0.004	−0.008, −0.001	0.126	−0.014	−0.004	−0.009, 0.001	0.089
4	−0.017	−0.004	−0.009, 0.000	0.037	−0.008	−0.002	−0.007, 0.002	0.314	−0.014	−0.004	−0.009, 0.001	0.096	−0.015	−0.005	−0.010, 0.000	0.058
5	0.000	0.000	−0.004, 0.004	0.959	0.009	0.002	−0.002, 0.007	0.274	0.001	0.000	−0.005, 0.005	0.920	−0.003	−0.001	−0.006, 0.004	0.718

Note: Variables in Model 1 = residential greenspace, unadjusted; Model 2 = Model 1 + age, sex; Model 3 = Model 2 + average neighborhood income; Model 4 = Model 3 + individual income; Model 5 = Model 4 + relationship status. Model 1 fit: 50 m *F*(1, 14 288) = 9.183, *p* = 0.002 adj. *R*^2^ = 0.001; 100 m *F*(1, 14 288) = 2.979, *p* = 0.084 adj. *R*^2^ = 0.000; 300 m *F*(1, 14 288) = 3.667, *p* = 0.056 adj. *R*^2^ = 0.000; 500 m *F*(1, 14 288) = 4.012, *p* = 0.045 adj. *R*^2^ = 0.000. Model 4 fit: 50 m *F*(5, 14 284) = 138.840, *p* < 0.001 adj. *R*^2^ = 0.046; 100 m *F*(5, 14 284) = 138.136, *p* < 0.001 adj. *R*^2^ = 0.046; 300 m *F*(5, 14 284) = 138.503, *p* < 0.001 adj. *R*^2^ = 0.046; 500 m *F*(5, 14 284) = 138.674, *p* < 0.001 adj. *R*^2^ = 0.046. B = unstandardized beta coefficient, β = standardized beta coefficient.

**Table 3 ijerph-19-05668-t003:** Associations between residential greenspace at 50 m, 100 m, 300 m, and 500 m buffers, and depressive symptoms (subsample: non-working).

	50 m Buffer				100 m Buffer				300 m Buffer				500 m Buffer			
Model, N = 4120	β	B	Lower, upper (95% CI of B)	*p*	β	B	Lower, upper (95% CI of B)	*p*	β	B	Lower, upper (95% CI of B)	*p*	β	B	Lower, upper (95% CI of B)	*p*
1	−0.056	−0.014	−0.022, −0.006	0.000	−0.048	−0.013	−0.021, −0.005	0.002	−0.041	−0.012	−0.021, −0.003	0.008	−0.040	−0.012	−0.021, −0.003	0.011
2	−0.038	−0.010	−0.017, −0.002	0.013	−0.032	−0.009	−0.017, −0.001	0.033	−0.029	−0.009	−0.017, 0.000	0.054	−0.028	−0.008	−0.017, 0.001	0.070
3	−0.040	−0.010	−0.018, −0.003	0.008	−0.034	−0.009	−0.017, −0.001	0.025	−0.030	−0.009	−0.017, 0.000	0.045	−0.030	−0.009	−0.018, 0.000	0.048
4	−0.042	−0.011	−0.018, −0.003	0.006	−0.037	−0.010	−0.018, −0.002	0.014	−0.036	−0.010	−0.019, −0.002	0.019	−0.036	−0.011	−0.020, −0.002	0.018
5	−0.024	−0.006	−0.014, 0.001	0.113	−0.019	−0.005	−0.013, 0.003	0.220	−0.020	−0.006	−0.014, 0.003	0.183	−0.023	−0.007	0.016, 0.002	0.135

Note: Variables in Model 1 = residential greenspace, unadjusted; Model 2 = Model 1 + age, sex; Model 3 = Model 2 + average neighborhood income; Model 4 = Model 3 + individual income; Model 5 = Model 4 + relationship status. Model 1 fit: 50 m *F*(1, 4 118) = 12.745, *p* < 0.001, adj. *R*^2^ = 0.003; 100 m *F*(1, 4 118) = 9.697, *p* = 0.002, adj. *R*^2^ = 0.002; 300 m *F*(1, 4 118) = 7.097, *p* = 0.008, adj. *R*^2^ = 0.001; 500 m *F*(1, 4 118) = 6.550, *p* = 0.011, adj. *R*^2^ = 0.001. Model 4 fit: 50 m *F*(5, 4 114) = 60.573, *p* < 0.001 adj. *R*^2^ = 0.067; 100 m *F*(5, 4 114) = 60.221, *p* < 0.001 adj. *R*^2^ = 0.067; 300 m *F*(5, 4 114) = 60.108, *p* < 0.001 adj. *R*^2^ = 0.067; 500 m *F*(5, 4 114) = 60.127, *p* < 0.001 adj. *R*^2^ = 0.067. B = unstandardized beta coefficient, β = standardized beta coefficient.

**Table 4 ijerph-19-05668-t004:** Associations between residential greenspace at 50 m, 100 m, 300 m, and 500 m buffers, and burnout symptoms (whole sample/population).

	50 m Buffer				100 m Buffer				300 m Buffer				500 m Buffer			
Model, N = 14,303	β	B	Lower, upper (95% CI of B)	*p*	β	B	Lower, upper (95% CI of B)	*p*	β	B	Lower, upper (95% CI of B)	*p*	β	B	Lower, upper (95% CI of B)	*p*
1	−0.019	−0.001	−0.002, 0.000	0.023	−0.009	−0.001	−0.002, 0.001	0.285	−0.009	−0.001	−0.002, 0.001	0.270	−0.011	−0.001	−0.002, 0.000	0.206
2	−0.008	−0.001	−0.002, 0.000	0.306	−0.001	0.000	−0.001, 0.001	0.917	−0.005	0.000	−0.002, 0.001	0.537	−0.007	−0.001	−0.002, 0.001	0.415
3	−0.010	−0.001	−0.002, 0.000	0.205	−0.002	0.000	−0.001, 0.001	0.822	−0.005	0.000	−0.002, 0.001	0.515	−0.008	−0.001	−0.002, 0.001	0.345
4	−0.010	−0.001	−0.002, 0.000	0.235	−0.002	0.000	−0.001, 0.001	0.824	−0.006	0.000	−0.002, 0.001	0.430	−0.009	−0.001	−0.002, 0.001	0.255
5	0.003	0.000	−0.001, 0.001	0.727	0.011	0.001	0.000, 0.002	0.174	0.004	0.000	−0.001, 0.002	0.589	0.000	0.000	−0.001, 0.001	0.991

Note: Variables in Model 1 = residential greenspace, unadjusted; Model 2 = Model 1 + age, sex; Model 3 = Model 2 + average neighborhood income; Model 4 = Model 3 + individual income; Model 5 = Model 4 + relationship status. Model 1 fit: 50 m *F*(1, 14 301) = 5.180, *p* = 0.023 adj. *R*^2^ = 0.000; 100 m *F*(1, 14 301) = 1.144, *p* = 0.285, adj. *R*^2^ = 0.000; 300 m *F*(1, 14 301) = 1.215, *p* = 0.270, adj. *R*^2^ = 0.000; 500 m *F*(1, 14 301) = 1.602, *p* = 0.206, adj. *R*^2^ = 0.000. Model 4 fit: 50 m *F*(5, 14 297) = 212.028, *p* < 0.001 adj. *R*^2^ = 0.069; 100 m *F*(1, 14 297) = 211.735, *p* < 0.001, adj. *R*^2^ = 0.069; 300 m *F*(1, 14 297) = 211.858, *p* < 0.001, adj. *R*^2^ = 0.069; 500 m *F*(1, 14 297) = 212.003, *p* < 0.001, adj. *R*^2^ = 0.069. B = unstandardized beta coefficient, β = standardized beta coefficient.

**Table 5 ijerph-19-05668-t005:** Associations between residential greenspace at 50 m, 100 m, 300 m, and 500 m buffers, and burnout symptoms (subsample: non-working).

	50 m Buffer				100 m Buffer				300 m Buffer				500 m Buffer			
Model, N = 4116	β	B	Lower, upper (95% CI of B)	*p*	β	B	Lower, upper (95% CI of B)	*p*	β	B	Lower, upper (95% CI of B)	*p*	β	B	Lower, upper (95% CI of B)	*p*
1	−0.046	−0.003	−0.005, 0.001	0.003	−0.040	−0.003	−0.005, 0.001	0.011	−0.042	−0.003	−0.005, 0.001	0.006	−0.041	−0.003	−0.005, 0.001	0.008
2	−0.023	−0.001	−0.003, 0.000	0.118	−0.019	−0.001	−0.003, 0.001	0.198	−0.027	−0.002	−0.004, 0.000	0.067	−0.026	−0.002	−0.004, 0.000	0.084
3	−0.025	−0.002	−0.003, 0.000	0.089	−0.020	−0.001	−0.003, 0.001	0.168	−0.028	−0.002	−0.004, 0.000	0.057	−0.028	−0.002	−0.004, 0.000	0.061
4	−0.026	−0.002	−0.003, 0.000	0.077	−0.023	−0.002	−0.003, 0.000	0.127	−0.032	−0.002	−0.004, 0.000	0.032	−0.032	−0.002	−0.005, 0.000	0.032
5	−0.016	−0.001	−0.003, 0.001	0.282	−0.012	−0.001	−0.003, 0.001	0.416	−0.023	−0.002	−0.004, 0.000	0.120	−0.024	−0.002	−0.004, 0.000	0.100

Note: Variables in Model 1 = residential greenspace, unadjusted; Model 2 = Model 1 + age, sex; Model 3 = Model 2 + average neighborhood income; Model 4 = Model 3 + individual income; Model 5 = Model 4 + relationship status. Model 1 fit: 50 m *F*(1, 4 114) = 8.684, *p* = 0.003, adj. *R*^2^ = 0.002; 100 m *F*(1, 4 114) = 6.470, *p* = 0.011, adj. *R*^2^ = 0.001; 300 m *F*(1, 4 114) = 7.433, *p* = 0.006, adj. *R*^2^ = 0.002; 500 m *F*(1, 4 114) = 6.999, *p* = 0.008, adj. *R*^2^ = 0.001. Model 4 fit: 50 m *F*(5, 4 110) = 99.003, *p* < 0.001 adj. *R*^2^ = 0.106; 100 m *F*(5, 4 110) = 98.825, *p* < 0.001 adj. *R*^2^ = 0.106; 300 m *F*(5, 4 110) = 99.336, *p* < 0.001 adj. *R*^2^ = 0.107; 500 m *F*(5, 4 110) = 99.340, *p* < 0.001 adj. *R*^2^ = 0.107. B = unstandardized beta coefficient, β = standardized beta coefficient.

**Table 6 ijerph-19-05668-t006:** Associations between residential greenspace at 50 m, 100 m, 300 m, and 500 m buffers, and life satisfaction (whole sample/population).

	50 m Buffer				100 m Buffer				300 m Buffer				500 m Buffer			
Model, N = 14,279	β	B	Lower, upper (95% CI of B)	*p*	β	B	Lower, upper (95% CI of B)	*p*	β	B	Lower, upper (95% CI of B)	*p*	β	B	Lower, upper (95% CI of B)	*p*
1	0.032	0.002	0.001, 0.003	0.000	0.025	0.002	0.001, 0.003	0.003	0.024	0.002	0.001, 0.003	0.004	0.018	0.001	0.000, 0.003	0.034
2	0.029	0.002	0.001, 0.003	0.000	0.023	0.002	0.000, 0.003	0.005	0.024	0.002	0.001, 0.003	0.004	0.018	0.001	0.000, 0.003	0.035
3	0.032	0.002	0.001, 0.003	0.000	0.025	0.002	0.001, 0.003	0.003	0.024	0.002	0.001, 0.003	0.003	0.019	0.001	0.000, 0.003	0.022
4	0.032	0.002	0.001, 0.003	0.000	0.025	0.002	0.001, 0.003	0.003	0.026	0.002	0.001, 0.003	0.002	0.021	0.002	0.000, 0.003	0.011
5	−0.003	0.000	−0.001, 0.001	0.748	−0.010	−0.001	−0.002, 0.000	0.206	−0.004	0.000	−0.001, 0.001	0.650	−0.005	0.000	−0.002, 0.001	0.563

Note: Variables in Model 1 = residential greenspace, unadjusted; Model 2 = Model 1 + age, sex; Model 3 = Model 2 + average neighborhood income; Model 4 = Model 3 + Individual Income; Model 5 = Model 4 + relationship status. Model 1 fit: 50 m *F*(1, 14 277) = 14.747, *p* < 0.001, adj. *R*^2^ = 0.001; 100 m *F*(1, 14 277) = 8.986, *p* = 0.003, adj. *R*^2^ = 0.001; 300 m *F*(1, 14 277) = 8.438, *p* = 0.004, adj. *R*^2^ = 0.001; 500 m *F*(1, 14 277) = 4.494, *p* = 0.034, adj. *R*^2^ = 0.000. Model 4 fit: 50 m *F*(5, 14 273) = 67.349, *p* < 0.001, adj. *R*^2^ = 0.023; 100 m *F*(5, 14 273) = 66.189, *p* < 0.001, adj. *R*^2^ = 0.022; 300 m *F*(5, 14 273) = 66.351, *p* < 0.001, adj. *R*^2^ = 0.022; 500 m *F*(5, 14 273) = 65.684, *p* < 0.001, adj. *R*^2^ = 0.022. B = unstandardized beta coefficient, β = standardized beta coefficient.

**Table 7 ijerph-19-05668-t007:** Associations between residential greenspace at 50 m, 100 m, 300 m, and 500 m buffers, and life satisfaction (subsample: non-working).

	50 m Buffer				100 m Buffer				300 m Buffer				500 m Buffer			
Model, N = 4102	β	B	Lower, upper (95% CI of B)	*p*	β	B	Lower, upper (95% CI of B)	*p*	β	B	Lower, upper (95% CI of B)	*p*	β	B	Lower, upper (95% CI of B)	*p*
1	0.052	0.003	0.001, 0.005	0.001	0.042	0.003	0.001, 0.005	0.007	0.037	0.003	0.000, 0.005	0.018	0.028	0.002	0.000, 0.005	0.072
2	0.046	0.003	0.001, 0.005	0.003	0.037	0.003	0.001, 0.005	0.018	0.034	0.003	0.000, 0.005	0.027	0.025	0.002	0.000, 0.004	0.103
3	0.049	0.003	0.001, 0.005	0.001	0.039	0.003	0.001, 0.005	0.012	0.036	0.003	0.000, 0.005	0.020	0.029	0.002	0.000, 0.005	0.063
4	0.051	0.003	0.001, 0.005	0.001	0.042	0.003	0.001, 0.005	0.006	0.041	0.003	0.001, 0.005	0.008	0.034	0.003	0.000, 0.005	0.026
5	0.021	0.001	−0.001, 0.003	0.162	0.012	0.001	−0.001, 0.003	0.447	0.016	0.001	−0.001, 0.003	0.306	0.013	0.001	−0.001, 0.003	0.405

Note: Variables in Model 1 = residential greenspace, unadjusted; Model 2 = Model 1 + age, sex; Model 3 = Model 2 + average neighborhood income; Model 4 = Model 3 + Individual Income; Model 5 = Model 4 + relationship status. Model 1 fit: 50 m *F*(1, 4 100) = 11.119, *p* = 0.001, adj. *R*^2^ = 0.002; 100 m *F*(1, 4 100) = 7.209, *p* = 0.007, adj. *R*^2^ = 0.002; 300 m *F*(1, 4 100) = 5.618, *p* = 0.018, adj. *R*^2^ = 0.001; 500 m *F*(1, 4 100) = 3.238, *p* = 0.072, adj. *R*^2^ = 0.001. Model 4 fit: 50 m *F*(5, 4 096) = 33.122, *p* < 0.001, adj. *R*^2^ = 0.038; 100 m *F*(5, 4 096) = 32.414, *p* < 0.001, adj. *R*^2^ = 0.037; 300 m *F*(5, 4 096) = 32.322, *p* < 0.001, adj. *R*^2^ = 0.037; 500 m *F*(5, 4 096) = 31.896, *p* < 0.001, adj. *R*^2^ = 0.036. B = unstandardized beta coefficient, β = standardized beta coefficient.

**Table 8 ijerph-19-05668-t008:** Associations between residential greenspace at 50 m, 100 m, 300 m, and 500 m buffers, and life satisfaction (subsample: working).

	50 m Buffer				100 m Buffer				300 m Buffer				500 m Buffer			
Model, N = 10,177	β	B	Lower, upper (95% CI of B)	*p*	β	B	Lower, upper (95% CI of B)	*p*	β	B	Lower, upper (95% CI of B)	*p*	β	B	Lower, upper (95% CI of B)	*p*
1	0.025	0.002	0.000, 0.003	0.012	0.019	0.001	0.000, 0.003	0.053	0.020	0.001	0.000, 0.003	0.043	0.014	0.001	0.000, 0.003	0.149
2	0.023	0.001	0.000, 0.003	0.019	0.018	0.001	0.000, 0.002	0.068	0.020	0.001	0.000, 0.003	0.044	0.014	0.001	0.000, 0.003	0.149
3	0.026	0.002	0.000, 0.003	0.008	0.019	0.001	0.000, 0.003	0.050	0.020	0.001	0.000, 0.003	0.042	0.015	0.001	0.000, 0.003	0.122
4	0.026	0.002	0.000, 0.003	0.008	0.019	0.001	0.000, 0.003	0.047	0.021	0.002	0.000, 0.003	0.032	0.017	0.001	0.000, 0.003	0.085
5	−0.010	−0.001	−0.002, 0.001	0.298	−0.018	−0.001	−0.002, 0.000	0.072	−0.010	−0.001	−0.002, 0.001	0.299	−0.010	−0.001	−0.002, 0.001	0.283

Note: Variables in Model 1 = residential greenspace, unadjusted; Model 2 = Model 1 + age, sex; Model 3 = Model 2 + average neighborhood income; Model 4 = Model 3 + Individual Income; Model 5 = Model 4 + relationship status. Model 1 fit: 50 m *F*(1, 10 175) = 6.359, *p* = 0.012, adj. *R*^2^ = 0.001; 100 m *F*(1, 10 175) = 3.747, *p* = 0.053, adj. *R*^2^ = 0.000; 300 m *F*(1, 10 175) = 4.111, *p* = 0.043, adj. *R*^2^ = 0.000; 500 m *F*(1, 10 175) = 2.087, *p* = 0.149, adj. *R*^2^ = 0.000. Model 4 fit: 50 m *F*(5, 10 171) = 37.803, *p* < 0.001, adj. *R*^2^ = 0.018; 100 m *F*(5, 10 171) = 37.163, *p* < 0.001, adj. *R*^2^ = 0.017; 300 m *F*(5, 10 171) = 37.304, *p* < 0.001, adj. *R*^2^ = 0.018; 500 m *F*(5, 10 171) = 36.967, *p* < 0.001, adj. *R*^2^ = 0.017. B = unstandardized beta coefficient, β = standardized beta coefficient beta coefficient.

## Data Availability

The data are not publicly available due to ethical and legal restrictions. For data requests, please contact the principal investigator and corresponding author of the study C.U.D.S.

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
