# Peer review of "Residential Greenspace Is Associated with Lower Levels of Depressive and Burnout Symptoms, and Higher Levels of Life Satisfaction: A Nationwide Population-Based Study in Sweden"

_ijerph, 2022, doi:10.3390/ijerph19095668_

Round 1

Reviewer 1 Report

Thank you for the opportunity to review this manuscript. In the study presented, the authors investigated whether residential greenspace and green-blue-space was associated with mental health outcomes in a large sample of Swedish adults. Greenspace and green-blue-space were identified using high resolution geographic data, and such space was stratified into 50, 100, 300, and 500 m buffer zones around the residential addresses of each participant. Each buffer zone was then used as a possible predictor variable for mental health outcomes, including depression, burnout, and life satisfaction. Hierarchical linear modelling was used to progressively control for additional variables such as age, sex, net income, and relationship status. On balance, a higher density of natural space in the closer buffer zones was associated with better mental health outcomes, especially in non-working individuals, until relationship status was accounted for.

The study is interesting, and is suitably motivated in a well-written introduction. However, some minor design issues and missing statistical parameters undermine the results and discussion presented. Below, some specific comments follow:

Methods:

  • The assessment of life satisfaction using only a single item seems particularly limiting. Little explanation is offered, other than mentioning that the item was “derived from the Whitehall II study”. No further comment is made about the possible limitations of this rudimentary measure of life satisfaction.

Results:

  • The authors do not mention any measure of reliability for the depression and burnout measures. Differences in measurement reliability might contribute to the different association strengths detected between natural space and mental health, thus these reliability measures would be of interest to readers.
  • A major issue regarding the linear regression results is the lack of model fit parameters. No model is presented with its corresponding F-statistic, degrees of freedom, p-value and (adjusted) R^2 value. This is problematic, because if a regression model does not have good fit to the observed data, then any discussion of apparently significant predictors is potentially meaningless. This is explained by Cohen, Cohen, West, and Aiken (2002, 3rd ed, p. 90):

“…if most variables do not account for more than a trivial amount of Y variance they may lower this average (the mean square for the regression) to the point of making the overall F not significant in spite of the apparent significance of the separate contributions of one or more individual IVs. In such circumstances, we recommend that such IVs not be accepted as significant. The reason for this is to avoid spuriously significant results, the probability of whose occurrence is controlled by the requirement that the F for a set of IVs be significant before its constituent IVs are t-tested.”

Without understanding the fit of each regression model, it is impossible to appreciate the significance of any single predictor. Moreover, without reporting the (adjusted) R2 value, the reader is provided no sense of the effect size of each model, and how this changes with each step in the hierarchical modelling. In the absence of these fit parameters, much of the discussion becomes moot, so I strongly suggest that the manuscript is revised to include these important statistics.

  • The authors do not mention if any casewise diagnostics were performed to assess the fit of their regression models? For example, were there any outlying observations, or observations that exerted significant influence on the regression line? Did all the models meet the assumptions of linear regression? In particular, the authors note a correlation between having a partner and a higher density of surrounding natural space. It would be informative to check the variance inflation factor of these two predictors in case of potential issues with collinearity.

Discussion:

  • There are several other limitations to the study that possibly merit some discussion. Firstly, the study instruments were all self-report, leading to potential issues such as context-effects and biased response sets (e.g. faking-good or faking-bad). Secondly, as noted earlier, there is no acknowledgement of the potential limitations of the instruments used (e.g. a single-item life satisfaction measure; all items measured in the same direction with no reverse-scored items). Thirdly, the season in which the survey was completed (spring) may have influenced responses, especially in a country such as Sweden which is well known for its lengthy winter periods.

Author Response

Dear Reviewer 1,

Thank you very much for reviewing our manuscript, and for your valuable and constructive feedback. We are very grateful for your critical comments and we are convinced that your input has improved the quality and clarity of our manuscript.

Comment 1: The assessment of life satisfaction using only a single item seems particularly limiting. Little explanation is offered, other than mentioning that the item was “derived from the Whitehall II study”. No further comment is made about the possible limitations of this rudimentary measure of life satisfaction.

Response: Thanks for pointing out potential limitations of using a single item life satisfaction measure, and that this may need further elaboration in the manuscript.

We think that it is important to also assess “positive” mental health outcomes, such as global life satisfaction, and not only symptoms of poor mental health. While single-item measures can limit the reliability, it has been found that life satisfaction measures with single items still have satisfactory reliability and validity, and show similar results as multiple-item measures. Especially items which are phrased in terms of general satisfaction with life (which is the case with the measure we are using) have been found to show the greatest reliability. See for example:

Diener, E., Inglehart, R., & Tay, L. (2013). Theory and validity of life satisfaction scales. Social indicators research, 112(3), 497-527.

Accordingly, we have now added more information and elaboration under 4.4 Strengths and limitations, and hope that we addressed your concern adequately (see the added text in the response to comment 5 further below).

Comment 2: The authors do not mention any measure of reliability for the depression and burnout measures. Differences in measurement reliability might contribute to the different association strengths detected between natural space and mental health, thus these reliability measures would be of interest to readers.

Response: Thank you for raising this point. We have now included the Cronbach’s  values for the depressive and burnout symptoms scales respectively. For both, Cronbach’s  are high and thus differences in the reliability of these scales do not explain differences in associations between RGS and these outcomes. We agree that informing the reader about the reliability of these scales is important.

Comment 3: A major issue regarding the linear regression results is the lack of model fit parameters. No model is presented with its corresponding F-statistic, degrees of freedom, p-value and (adjusted) R^2 value. This is problematic, because if a regression model does not have good fit to the observed data, then any discussion of apparently significant predictors is potentially meaningless. This is explained by Cohen, Cohen, West, and Aiken (2002, 3rd ed, p. 90):

“…if most variables do not account for more than a trivial amount of Y variance they may lower this average (the mean square for the regression) to the point of making the overall F not significant in spite of the apparent significance of the separate contributions of one or more individual IVs. In such circumstances, we recommend that such IVs not be accepted as significant. The reason for this is to avoid spuriously significant results, the probability of whose occurrence is controlled by the requirement that the F for a set of IVs be significant before its constituent IVs are t-tested.”

Without understanding the fit of each regression model, it is impossible to appreciate the significance of any single predictor. Moreover, without reporting the (adjusted) R2 value, the reader is provided no sense of the effect size of each model, and how this changes with each step in the hierarchical modelling. In the absence of these fit parameters, much of the discussion becomes moot, so I strongly suggest that the manuscript is revised to include these important statistics.

Response: Thank you for pointing this out. We have now included all the recommended model fit statistics of the different buffer zones in the unadjusted models and models after step 4, within each of the tables. We agree that these parameters are important for the interpretation of results, and that reporting them improves clarity.

Comment 4: The authors do not mention if any casewise diagnostics were performed to assess the fit of their regression models? For example, were there any outlying observations, or observations that exerted significant influence on the regression line? Did all the models meet the assumptions of linear regression? In particular, the authors note a correlation between having a partner and a higher density of surrounding natural space. It would be informative to check the variance inflation factor of these two predictors in case of potential issues with collinearity.

Response: Our models met the assumptions (linearity, approximate normality, homoscedasticity, no multicollinearity) for linear regression. Outliers were not present, due to the nature of the measures used (i.e. the health measures have standardized rating scales with a minimum and maximum score, and demographic control variables also have clear min-max scores or categories). Furthermore, the possibility of nonlinearity, i.e., quadratic or cubic trends in associations between greenspace and outcomes, was checked, but no support for non-linearity was found. No superiority of quadratic or cubic model fit over linear models was observed. With respect to normality, it is important to consider the central limit theorem, given the large sample size. Moreover, we have analyzed the Durbin-Watson statistic (constantly close to 2), inter-item correlations (also between relationship status and greenspace variables (between around 0.13 - 0.17)) the tolerance (constantly far above 0.1, and mostly around 0.9), and the variance inflation factor (constantly around 1, i.e., far from 10).

To clarify that the assumptions of linear regression were met we have now added the following sentence to 2.3 Statistical analysis:

“The assumptions for linear regression were met (e.g. Tabachnick & Fidell, 2007 [45])”.

Comment 5: There are several other limitations to the study that possibly merit some discussion. Firstly, the study instruments were all self-report, leading to potential issues such as context-effects and biased response sets (e.g. faking-good or faking-bad). Secondly, as noted earlier, there is no acknowledgement of the potential limitations of the instruments used (e.g. a single-item life satisfaction measure; all items measured in the same direction with no reverse-scored items). Thirdly, the season in which the survey was completed (spring) may have influenced responses, especially in a country such as Sweden which is well known for its lengthy winter periods.

Response: Concerning the discussion of the suggested limitations, we have now added a discussion elaborating on the strengths and limitations of self-report measures and the single-item life satisfaction measure (as mentioned above) in the section 4.4 Strengths and limitations:

“However, the use of self-report health scales can have both advantages and disadvantages. One possible disadvantage is that they may be subject to biased response sets (i.e., over- or under-reporting symptoms) or context effects. On the other hand, such biases in symptom reporting apply also in clinical settings and assessments. The major advantages of the self-report measures used in the present research are that they allow the investigation of symptom levels, which was the aim of the present study. That is, the full range of symptoms from none to severe are assessed, thus including sub-clinical symptoms, and not merely e.g., a registered diagnosis. Furthermore, mental health is inherently about the individual experience (of positive and negative symptoms), whether reported in a de-identified survey or to a health care practitioner. It has also been found that people are good at rating their own general health, where such self-ratings are predictive of a range of health outcomes (all-cause mortality, future functional status, treatment outcomes), and more so than some “objective” health measures like diagnoses [55]. Furthermore, in the present study, fine-grained individual objective environmental and demographic data were studied in relation to individual health measures, as such preventing risks of common method bias which can be an issue when both predictors and outcomes are self-rated [56]. However, further complementary studies of other clinical outcomes would be valuable. Generally, self-report measures with multiple items, such as the depression and burnout scales, yield better scale properties [57]. Here, the use of a single-item measure of global life satisfaction may be considered a limitation. However, single-item measures of life satisfaction have been found reliable, robust, and valid, for instance in terms of test-retest reliability and in producing similar results to those of multi-item scales [58], while minimizing the respondent burden, reducing criterion contamination, or improving face validity [57,59]. Furthermore, items asking about general satisfaction with life (as used in the present study) have been found most reliable. This motivates the use of a single-item life satisfaction measure, which has also been used and validated before [39,40], to include a positive measure of mental health.”

Regarding seasonal effects on mental health symptoms, these become more relevant when having multiple measurement points over the year. The present study has a cross-sectional design where the survey is conducted nationwide during the spring, and thus the seasonal effect on the health outcomes should be similar across the study sample regardless of greenspace levels.

The time point of the data collection in the study cohort has been intentionally planned to be during springtime to minimize potential seasonal effects, as may occur during the winter season on mental well-being. One could speculate that associations between greenspace and health outcomes may be even stronger after summer, due to the summer being a time of greener and flowering environments, that people may spend more time outdoors in the residential surrounding, and that these aspects enhance the benefits of residential greenspace even more if assessed after the summer. As such, if anything, the observed effects could be underestimated in the present study. However, one should also bear in mind that assessment of mental health outcome levels after the summer, which is the main holiday season, could also introduce other types of effects on mental health outcomes that are independent of residential greenspace since the holiday season is a time when people may also go away and instead get a boost in mental well-being due to holidaying (e.g. due to getting rest from work, spending time in alternative nice locations and getting a change of scene, doing recuperative and inspiring activities).

We hope that we have addressed your comments to your satisfaction and thank you again for your valuable input.

Best regards,

The corresponding authors, on behalf of all authors

Reviewer 2 Report

In the current study authors aimed at investigating the role of individual Red green space in mental health, in a population based sample of adults in Swedish urban areas. For this the authors proposed and proved three hypotheses:

  • More RGS is associated with better mental health outcomes.
  • RGS in immediate residence surrounding environment play the greatest role for mental health outcomes.
  • Mental health outcomes are stronger inn combined green and blue spaces versus green spaces in isolation.

Mental health outcomes were assessed based on depressive and burnout symptoms and life satisfaction. Although somewhat obvious this study adds to the scientific evidence that existence of greenspace in immediate residential area supports mental health in the population.

Overall, the current study is appropriately designed and conducted. The results are in line with conclusions. I support the publication of this manuscript.  

I have read a few articles where authors worked in similar directions but in different populations. Below are a few example.

https://www.pnas.org/doi/10.1073/pnas.1807504116

https://www.ncbi.nlm.nih.gov/pmc/articles/PMC7557737/

https://bmcpsychiatry.biomedcentral.com/articles/10.1186/s12888-018-1926-1

https://www.sciencedirect.com/science/article/pii/S0013935122003838

Overall these studies have resulted in mixed association between green spaces and mental health. This could be due to no account for the actual individual exposure to greenness, inconsistencies in defining green spaces, Sample size etc. In the current study however, the authors made an effort to investigate the role of individual green space (this encompasses both public and private spaces) at different Euclidean residential buffer zones for mental health outcomes.

This is a significant step forward in the field. The experimental design is appropriate. Although I would have like to see the effect of just the blue spaces on mental health and if they are comparable to green space or green+blue space combined.

The conclusions are in line with the results. Overall it's a well designed (and written) coherent work which aims at increasing the quality of life.

Author Response

Dear Reviewer 2,

Thank you very much for reviewing our manuscript, and for your valuable and supportive feedback. We are very grateful for your comments and for your acknowledgment of our work.

Best regards,

The corresponding authors, on behalf of all authors

Reviewer 3 Report

Thank you for giving me this opportunity to read the manuscript entitled "Residential Greenspace is Associated with Lower Levels of Depressive and Burnout Symptoms, and Higher Levels of Life  Satisfaction: A Nationwide Population-Based Study in Sweden". The topic of this manuscript is interesting and would be a good contribution to this field. I think it could be considered for publication in International Journal of Environmental Research and Public Health once the following issues are addressed.

  1. Please replace the keywords that already appear in the manuscript’s title with close synonyms or other keywords, which will also facilitate your paper to be searched by potential readers.

  1. Lines 126-130: The method (i.e., satellite image classification model) for greenspace and blue space mapping should be specified here. A map showing the greenspace (and blue space) of the study area will help readers to understand how the key variables were quantified in this study.

  1. As far as I know, 500 m, 1000 m and 1500 m are commonly used buffers for environmental exposure assessment. I would be very interested to know the reason why the authors chose such close scales (namely 50 m &100 m, 300 m & 500 m), as they do not have significant differences in quantifying green space exposure levels and revealing environmental health effects.

  1. Table 1 “Count (%) Mean individual income in 100’s SEK (SD)”: I think the (SD) here is unnecessary.

  1. Human mobility is not considered in the greenspace (and blue space) exposure assessment, which will inevitably create uncertainty. This issues should be discussed in the Limitation section and the following papers are suggested to be cited as references: (1) “Dynamic assessments of population exposure to urban greenspace using multi-source big data”, and (2) “Observed inequality in urban greenspace exposure in China”.

  1. Some grammatical errors exist in the manuscript. Therefore, a critical review of the manuscript language will improve readability.

Author Response

Dear Reviewer 3,

Thank you very much for reviewing our manuscript, and for your valuable feedback. We are very grateful for your critical comments and are convinced that your input has improved the quality and clarity of our manuscript.

Comment 1: Please replace the keywords that already appear in the manuscript’s title with close synonyms or other keywords, which will also facilitate your paper to be searched by potential readers.

Response: We have now replaced the keywords that already appear in the title.

Comment 2: Lines 126-130: The method (i.e., satellite image classification model) for greenspace and blue space mapping should be specified here. A map showing the greenspace (and blue space) of the study area will help readers to understand how the key variables were quantified in this study.

Response: All our data (including satellite images) are collected nationwide, across Sweden. Therefore, it is unfortunately not possible to include a “map showing the greenspace (and blue space) of the study area. However, we refer to the Statistics Sweden, 2019, report on greenspace and green areas in urban areas 2015, where more information (including visualizations) can be found. Moreover, we have further clarified the method and have now added an example map of general greenspace for one of the greenest urban areas in Sweden in the Supplement (i.e., Lindingö, one of the top greenest urban areas in Sweden), which we now also refer to in the Methods section.

Comment 3: As far as I know, 500 m, 1000 m and 1500 m are commonly used buffers for environmental exposure assessment. I would be very interested to know the reason why the authors chose such close scales (namely 50 m &100 m, 300 m & 500 m), as they do not have significant differences in quantifying green space exposure levels and revealing environmental health effects.

Response: The focus of the present study is to investigate “closer” buffer zones in relation to mental health, as those have been recommended in prior research for the investigation of mental health outcomes, and for investigating more “automatic” residential greenspace exposure, which is less dependent on the individuals’ physical activity and mobility (i.e. exposure depends on the individual going further away from the home to reach the green area). E.g.:

S.M. Labib, S. Lindley, J.J. Huck, Spatial dimensions of the influence of urban green-blue spaces on human health: A systematic review, Environ. Res. 180 (2020) 108869. https://doi.org/10.1016/j.envres.2019.108869.

We have also outlined the focus on the environment in the close residential surroundings, and more automatic residential exposures in the introduction, and have clarified this further (addition is underlined):

“greenspace has often been assessed at a non-specific aggregate level such as general greenness at the neighborhood, district, or even city level. This results in uncertainty of the actual individual exposure to greenness and limits the accuracy of exposure as well as the conclusions which can be drawn from study results (see e.g., Labib et al., 2020 [24], where related methodological aspects are further discussed, including the Modifiable Areal Unit Problem). The inconsistencies in prior study results may thus partly relate to methodological issues, including the use of greenspace exposure measures at a crude level and not at the individual address level, only accounting for publicly available greenspace and not all greenspace, and/or studying limited/small samples. Consequently, there is a need for population-based studies investigating higher quality greenspace measures at the individual level. That is, there is a need for population-based studies assessing the role of greenspace in the more immediate residential surrounding environment which individuals are more automatically exposed to on a daily basis, which are more independent of mobility and physical activity levels (e.g. Labib et al 2020; Markevych et al. 2017 [24,25]).”

Comment 4: Table 1 “Count (%) Mean individual income in 100’s SEK (SD)”: I think the (SD) here is unnecessary.

Response: Thank you for this observation and for pointing this out. This has now been corrected.

Comment 5: Human mobility is not considered in the greenspace (and blue space) exposure assessment, which will inevitably create uncertainty. This issues should be discussed in the Limitation section and the following papers are suggested to be cited as references: (1) “Dynamic assessments of population exposure to urban greenspace using multi-source big data”, and (2) “Observed inequality in urban greenspace exposure in China”.

Response: This is a very relevant point and human mobility in relation to greenspace is an important research area. The methods used in the suggested articles are very interesting and valuable for addressing these aspects.  However, in the present study, we do not have access to such data. This point taps into what we also write about under comment 3, that the present study focuses on the more automatic exposure in individuals’ residential living environment and focuses on “closer” buffer zones. We very much agree, however, that human mobility-dependent greenspace exposure needs to be investigated further in future research. Thus, we have added a comment on this in the strengths and limitations section, with reference also to Song et al. 2018:

“The present study focused on the role of individual residential greenspace exposure in the closer residential surrounding—exposure which is more automatic and less dependent on the individual’s mobility to reach green areas further away from the residence. However, this can also be regarded as a limitation. Non-residential activity- and mobility-dependent greenspace exposure in relation to health outcomes in population-based samples is an important area to study further, where examples of such methodologies already exist (see e.g. Labib et al. 2020 and Song et al. 2018).”

Comment 6: Some grammatical errors exist in the manuscript. Therefore, a critical review of the manuscript language will improve readability.

Thank you for pointing this out. We have addressed this and the manuscript has been screened and any grammatical errors have been corrected.

We hope that we have addressed your comments to your satisfaction and thank you again for your valuable input.

Best regards,

The corresponding authors, on behalf of all authors

Round 2

Reviewer 1 Report

My sincere thanks to the authors for comprehensively addressing the comments from my initial review. The addition of the instrument reliability and regression model fit parameters improves the clarity of the results presented, and the revised strengths and limitations subsection better frames the overall impact of the study. I recommend the manuscript is accepted for publication.  

As a minor aside, I do note, however, that the newly added adjusted R2 values indicate that all regression models are of limited effect size - a limitation that the authors may also wish to acknowledge in the final revision of their paper. 

Reviewer 3 Report

Thank you for giving me this opportunity to read the revised version of the manuscript titled "Residential Greenspace is Associated with Lower Levels of Depressive and Burnout Symptoms, and Higher Levels of Life Satisfaction: A Nationwide Population-Based Study in Sweden", and for the detailed responses to my earlier comments. I am satisfied with this revised version, and I think it is acceptable now.